# Birth prevalence and parental stress associated with neural tube defects in Amhara's public comprehensive specialized hospitals, Ethiopia, 2024

Hailemariam Gezie[1], Endalk Birrie Wondifraw[2], Muluken Amare Wudu[2], Habtam Gelaye[3], Fekadeselassie Belege Getaneh[2]*

1 Department of Emergency and Critical Care Nursing, College of Medicine and Health Sciences, Wollo University, Dessie, Ethiopia, 2 Department of Pediatrics and Child Health Nursing, College of Health Sciences, Wollo University, Dessie, Ethiopia, 3 Department of Psychiatry, College of Health Sciences, Wollo University, Dessie, Ethiopia

* fekadebelege@gmail.com, fekadeselassie.belege@wu.edu.et

## Abstract

### Background

Neural tube defects (NTDs) are severe congenital anomalies resulting from the incomplete closure of the embryonic neural tube, affecting around 300,000 new-borns globally each year and leading to significant mortality and disability. While high-income countries have seen a reduction in NTD prevalence, developing nations like Ethiopia continue to face high rates. Families impacted by NTDs often endure emotional challenges, including grief, anxiety, and social isolation. This study aims to investigate the birth prevalence of NTDs and the associated parental stress, empha-sizing the wider effects on families.

### Methodology

An institution-based cross-sectional study was conducted in Dessie and Deber Berhan comprehensive specialized hospitals from July 24, 2023, to July 24, 2024, to evaluate the birth prevalence of NTDs and the associated parental stress among parents of children aged 1 month to 12 years diagnosed with NTDs. A total of 308 parent-child pairs participated in the study. Data were gathered using a pretested questionnaire and an 18-item Parenting Stress Scale. Statistical analysis was performed using Stata version 17, where linear regression was utilized to identify significant predictors after verifying the necessary assumptions. The findings were presented in multiple formats for clarity and comprehensibility.

**Data availability statement:** All relevant data are within the paper and its Supporting Information files.

**Funding:** This study was supported by Wollo University. The funder had no role in the study design, data collection, data analysis, data interpretation, or the writing of the manuscript.

**Competing interests:** The authors have declared that no competing interests exist.

## Results

The overall birth prevalence of neural tube defects was found to be 0.0052 (95% CI: 0.0038, 0.0067), which translates to 52 cases per 10,000 deliveries. Key factors associated with increased parental stress included being a mother ($\beta = 2.51$), older parental age ($\beta = 0.18$), the child's age ($\beta = 0.81$), a prior history of having children with NTDs ($\beta = 7.88$), and the presence of a ventriculoperitoneal shunt in the child ($\beta = 4.66$).

## Conclusion

The findings of this study indicate that the birth prevalence of NTDs is becoming a significant public health concern. Additionally, several factors contributing to increased parental stress were identified, including older parental age, the child's age, a previous history of NTDs in siblings, and the presence of a ventriculoperitoneal shunt. These results highlight the urgent need for targeted support and resources for affected families to help mitigate the psychological impact associated with these conditions.

## Introduction

Neural tube defects (NTDs) are significant congenital anomalies affecting the brain and spinal cord, resulting from the incomplete closure of the embryonic neural tube during early pregnancy [1]. Globally, approximately 300,000 babies are born with NTDs each year, leading to around 88,000 deaths and 8.6 million disability-adjusted life years (DALYs) [2].While high-income countries have experienced a decline in both the birth and total prevalence of NTDs over the past three decades, the situation remains critical in developing nations, where the incidence can reach up to 130 per 10,000 births [3].

In low-income countries, NTDs account for about 29% of neonatal deaths due to observable birth defects [2]. In Ethiopia, the incidence of NTDs has alarmingly increased, with a pooled birth prevalence of 61.43 per 10,000 [1], and some studies report rates as high as 131 cases per 10,000 children [4]. Maternal folate deficiency is a major contributing factor to the prevalence of NTDs; however, many families remain unaware of its importance. Evidence suggests that folic acid supplementation can reduce NTD incidence by 30–50% [5–7].

The consequences of NTDs extend beyond medical complications, as they impose considerable psychological and emotional burdens on affected families.[8,9]. Parents facing an NTD diagnosis often experience profound grief, anxiety, and stress [10], along with heightened distress, social isolation, and challenges in coping with their child's condition [11–13]. Recent studies indicate that parents of children with NTDs experience significant psychological impacts, including increased levels of stress and anxiety [14,15].

In Ethiopia, where healthcare resources are limited, the psychological and financial strains are exacerbated. This study aims to determine the birth prevalence of

NTDs and assess parental stress among caregivers of children attending follow-up clinics in the Amhara regional state. By identifying factors associated with elevated stress in this vulnerable population, the research seeks to inform the development of timely, context-appropriate psychosocial support systems. Addressing these gaps is essential to advancing comprehensive, family-centered care that responds not only to the clinical needs of children with NTDs but also to the dynamic emotional journey of their parents.

## Methods

### Study area and period

The study was conducted at Dessie Comprehensive Specialized Hospital (DCSH) and Debre Berhan Comprehensive Specialized Hospital (DBCSH) from July 24, 2023, to July 24, 2024. Both hospitals are tertiary-level public institutions located in the Eastern Amhara region of Ethiopia and serve as primary referral centers for high-risk pregnancies and complex neonatal conditions from surrounding zones, including North Wollo, South Wollo, Oromia, and parts of Afar. As such, they manage a disproportionate share of complicated deliveries in the region, including those referred for suspected or confirmed congenital anomalies. While DCSH and DBCSH together account for a substantial portion of institutional deliveries in their catchment areas, the majority of births in the broader Amhara region still occur at home or in lower-level health facilities. This referral bias is an important consideration when interpreting the observed birth prevalence of neural tube defects, as our institution-based sample likely overrepresents severe or diagnosed cases compared to the general population.

### Study design

An institution-based cross-sectional study was conducted to evaluate the birth prevalence and stress levels among parents of babies with neural tube defects who visited selected hospitals for follow-up care.

### Populations

The source population included all mothers who gave birth in the selected public comprehensive specialized hospitals in Eastern Amhara for estimating birth prevalence. It also included all parent-child pairs who had follow-up visits at these hospitals to assess parental stress. The study population specifically comprised mothers who gave birth in the selected hospitals and parents of children aged 1 month to 12 years diagnosed with NTDs who attended follow-up clinics during the study period.

### Eligibility criteria

Inclusion criteria encompassed mothers who gave birth in the selected hospitals and parents of children with NTDs attending follow-up clinics. Exclusion criteria included parents who were unable to communicate due to mental illness or those who were accompanied by relatives during the interviews.

### Sample size and sampling procedure

To assess the birth prevalence of neural tube defects and the psychological stress among parents of affected infants, we used a birth prevalence rate of 1.09% [16]. This resulted in a required sample size of 17 participants. Additionally, a study conducted in Addis Ababa revealed that 17.6% [17] of parents experience stress. Based on this information, the estimated sample size is 223. After incorporating a 10% non-response rate, the final sample size was adjusted to 245.

However, to improve the study's precision, we collected data on all birth outcomes between July 24, 2023, and July 24, 2024, at two selected comprehensive specialized hospitals in East Amhara to determine the birth prevalence of NTD. We also included parents who visited follow-up clinics during this time to assess the psychological stress related to their child's condition and the factors contributing to it.

### Study variables

**Dependent variables.** Presence of NTDs at birth (Yes/No) and parental stress related to having a child with NTDs.
**Independent variables.** Parental age, income, family size, marital status, residency, child's age and sex, type of NTD, ability to ambulate, urinary incontinence, bowel dysfunction, Surgical management, presence of Shunt, physiotherapy, and clean intermittent catheter.

### Operational definitions

- **Birth Prevalence of NTDs:** The number of live births or stillbirths with a confirmed diagnosis of neural tube defect (NTD) occurring at or after 28 weeks of gestation, divided by the total number of births (live births + stillbirths ≥28 weeks) during the study period. NTD diagnoses were established through physical examination by attending obstetricians or residents at birth; in cases of obvious external anomalies (e.g., anencephaly, myelomeningocele), diagnoses were initially identified by trained midwives and subsequently confirmed by physicians.

- **Parental stress**: Emotional and psychological responses of parents related to their child's condition. It was assessed using an 18-item Parenting Stress Scale.

### Data collection tool and Data collection procedure

Two trained Bachelor of Science (BSC) nurses collected data using structured and pretested tools and face-to-face interviews. The data collection instrument included socio-demographic, neonatal, and obstetrical characteristics. To assess parental stress levels regarding their infants with NTDs, an 18-item short-form Parental Stress Scale (PSS) was employed. This scale, developed in 1995 and revised for cultural relevance, targets parents of children aged 1 month to 12 years [18]. It includes ten negative and eight positive statements, with responses rated on a 5-point Likert scale. A sum score is calculated, where lower scores indicate less stress and higher scores indicate more. The questionnaire was translated into Amharic and back-translated to ensure accuracy and clarity.

### Data quality assurance

Training was provided for data collectors and supervisors, and a pretest was conducted to ensure the clarity and reliability of the questionnaire (Cronbach's $\alpha = 0.826$). Data completeness and consistency were monitored throughout the collection period.

### Data processing and analysis

Data processing and analysis were conducted using Epi Data version 3.1 and Stata version 17.0 software. Descriptive statistics, including means (SD) or medians (IQR) for continuous variables and frequencies and percentages for categorical variables, were used to summarize the participants' baseline characteristics.

Before regression analysis, assumptions such as linearity, independence of observations, normality (assessed via the Shapiro-Wilk test: $W = 0.992$, $p = 0.089$), homoscedasticity (evaluated using the Breusch-Pagan/Cook-Weisberg test, $p = 0.963$), and multicollinearity ($VIF = 1.45$) were verified. After performing a simple linear regression, variables with p-values less than 0.25 were included in subsequent multiple linear regression analyses. Statistically significant predictors were identified at a significance level of $p < 0.05$. The results were organized and presented in text and tables.

### Ethics approval and consent to participate

Ethical clearance for the study was granted by the Institutional Research Review Committee (IRC) of Wollo University, College of Medicine and Health Sciences, chaired by Mr. Mengistu Abate (Reference No. WU/0036/2023). Before data collection, a

permission letter was also obtained from the quality teams of each participating hospital. Participation in the study was voluntary, and written informed consent was secured from all eligible participants, ensuring adherence to ethical guidelines, including the Declaration of Helsinki. Participant confidentiality was maintained through the use of unique identification codes.

## Results

During the study period, 308 parents visited the selected hospital with their babies for follow-up, comprising 198 from DCSH and 110 from DBCSH. Among these parents, 245 (79.6%) were mothers. The median age of the parents was 29 years (IQR: 8 years), with ages ranging from 20 to 50 years. The average income of participants was 5,000 Ethiopian birrs (ETB). Most participants were married (90.9%) and resided in urban areas (77.3%) **"Table 1"**.

Of the 308 children with neural tube defects who visited follow-up clinics, over half (56.8%) were male. The mean age was 2.6 years (SD: 2.5 years). More than 80% (252) were diagnosed with myelomeningocele. Associated complications included inability to ambulate (70.1%), urinary incontinence (38.3%), and bowel dysfunction (37.3%). The majority (93.5%) of the children received surgical management, with 67 (22%) undergoing ventriculoperitoneal shunt procedures. Additionally, one-fourth of the children accessed physiotherapy services **"Table 2"**.

### Birth prevalence of neural tube defects

During the study period, a total of 9,734 deliveries occurred in the participating hospitals (5,869 at DCSH and 3,865 at DBCSH). Among these, 51 neural tube defect (NTD) cases were identified, yielding an overall birth prevalence of 0.0052

**Table 1. Socio-demographic characteristics of parents of a child with neural tube defect who had a follow-up in DCSH and DBCSH, Ethiopia 2024 (n = 308).**

| Characteristics | Category | Frequency | Percentage |
|---|---|---|---|
| Parental role | Mother | 245 | 79.55 |
| | Father | 63 | 20.45 |
| Age of the parents (in years) | Median (IQR) | 29 (8) | |
| Average monthly income (ETB) | Median (IQR) | 5000 (5000) | |
| Marital status | Married | 280 | 90.91 |
| | Unmarried | 28 | 9.09 |
| Residency | Urban | 238 | 77.27 |
| | Rural | 70 | 22.73 |
| Educational status of the mother | No formal education | 54 | 17.53 |
| | Primary education | 95 | 30.84 |
| | High school education | 70 | 22.73 |
| | Diploma and above | 89 | 28.90 |
| Educational status of the father | No formal education | 35 | 11.36 |
| | Primary education | 78 | 25.32 |
| | High school education | 72 | 23.38 |
| | Diploma and above | 123 | 39.94 |
| Occupational status of the respondent | Governmental employee | 84 | 27.27 |
| | Private employee | 90 | 29.22 |
| | Housewife | 126 | 40.91 |
| | Other | 8 | 2.6 |
| Number of families in the household | Four and less | 224 | 72.73 |
| | Five and above | 84 | 27.27 |

Table 2. Medical history and management-related characteristics of children with neural tube defect who have a follow-up in DCSH and DBCSH, Ethiopia 2024 (n = 308).

| Characteristics | Category | Frequency | Percentage |
|---|---|---|---|
| Sex of the child | Male | 175 | 56.82 |
| | Female | 133 | 43.18 |
| Age of child (in Years) | Mean (Sd) | 2.6 (± 2.5) | |
| Types of neural tube defect | Meningocele | 42 | 13.64 |
| | Myelomeningocele | 252 | 81.82 |
| | Encephalocele | 14 | 4.55 |
| Able to ambulate | Yes | 92 | 29.87 |
| | No | 216 | 70.13 |
| Urinary incontinency | Yes | 118 | 38.31 |
| | No | 190 | 61.69 |
| Bowel dysfunction | Yes | 115 | 37.34 |
| | No | 193 | 62.66 |
| Surgical management | Yes | 288 | 93.51 |
| | No | 20 | 6.49 |
| Ventricle-peritoneal Shunt | Yes | 67 | 21.75 |
| | No | 241 | 78.25 |
| Physiotherapy | Yes | 79 | 25.65 |
| | No | 229 | 74.35 |
| Clean intermittent catheter | Yes | 36 | 11.69 |
| | No | 272 | 88.31 |

(95% CI: 0.0038, 0.0067), or 52 per 10,000 deliveries. Of the 51 NTD cases, 34 were live births and 17 were stillbirths. The birth prevalence of NTDs differed slightly between the two hospitals: 32 cases were identified among 5,869 deliveries at DCSH (0.55%), and 19 cases among 3,865 deliveries at DBCSH (0.49%).

The distribution of NTD subtypes was as follows: spina bifida (including myelomeningocele, meningocele, and spina bifida occulta) accounted for 49.1% (n = 25), anencephaly for 31.4% (n = 16), encephalocele for 11.7% (n = 6), and multiple/complex defects for 7.8% (n = 4). All 16 anencephaly cases and one case of multiple defects resulted in stillbirth, consistent with the known lethality of these conditions.

## Stress among parents with neural tube defect children

The results of the 18-item self-report Parental Stress Scale reveal varied levels of parental stress and satisfaction among respondents. A significant number (89.9%) of parents may experience dissatisfaction in their role as a parent. Regarding the commitment to their children, 32% of respondents disagreed with the statement that they would do anything for their child(ren) if necessary. Thirty-seven percent of parents expressed agreement (combining strong agreement and agreement) with the statement of feeling close to their child(ren). Furthermore, 24.4% felt overwhelmed by the responsibilities of parenting, "**Table 3**".

## Factors associated with parental stress

In the simple linear regression analysis, variables with a P-value greater than 0.25 (such as marital status, family size, child sex, surgical therapy, and physiotherapy) were excluded from further analysis. The multiple linear regression analysis indicated that several factors were significantly associated with parental stress (P-value < 0.05), including being a mother, parental age, previous history of a child with neural tube defects, child's age, and the presence of a shunt.

**Table 3. Stress on parents with neural tube defect children who have followed up in DCSH and DBCSH, Ethiopia 2024 (n = 308).**

| 18-item self-report Parental Stress Scale | Strongly agree | Agree | Undecided | Disagree | Strongly disagree |
|---|---|---|---|---|---|
| | Freq. (%) | Freq. (%) | Freq. (%) | Freq. (%) | Freq. (%) |
| I am happy in my role as a parent* | 5 (1.6) | 14 (4.6) | 12 (3.9) | 204 (66.2) | 73 (23.7) |
| There is little or nothing I wouldn't do for my child(ren) if it was necessary* | 15 (4.9) | 89 (28.9) | 19 (6.2) | 99 (32.1) | 86 (27.9) |
| Caring for my child(ren) sometimes takes more time and energy than I have to give | 99 (32.1) | 126 (40.9) | 16 (5.2) | 63 (20.5) | 4 (1.3) |
| I sometimes worry whether I am doing enough for my child(ren) | 29 (9.4) | 55 (17.9) | 28 (9.1) | 186 (60.4) | 10 (3.2) |
| I feel close to my child(ren)* | 64 (20.8) | 51 (16.6) | 158 (51.3) | 19 (6.2) | 16 (5.1) |
| I enjoy spending time with my child(ren)* | 7 (2.3) | 39 (12.7) | 17 (5.5) | 179 (58.1) | 66 (21.4) |
| My child(ren) is an important source of affection for me* | 12 (3.9) | 97 (31.5) | 10 (30.3) | 110 (35.7) | 79 (25.6) |
| Having child(ren) gives me a more certain and optimistic view for the future* | 15 (4.9) | 79 (25.7) | 18 (5.8) | 106 (34.4) | 90 (29.2) |
| The major source of stress in my life is my child(ren) | 82 (26.6) | 112 (36.4) | 12 (3.9) | 77 (25) | 25 (8.1) |
| Having child(ren) leaves little time and flexibility in my life | 32 (10.4) | 72 (23.4) | 2 (0.7) | 195 (63.3) | 7 (2.2) |
| Having child(ren) has been a financial burden | 85 (27.6) | 119 (38.6) | 19 (6.2) | 73 (23.7) | 12 (3.9) |
| It is difficult to balance different responsibilities because of my child(ren) | 41 (13.3) | 88 (28.6) | 10 (3.3) | 155 (50.3) | 14 (4.5) |
| The behavior of my child(ren) is often embarrassing or stressful to me | 32 (10.4) | 171 (55.5) | 20 (6.5) | 74 (24) | 11 (3.6) |
| If I had it to do over again, I might decide not to have child(ren) | 55 (17.9) | 53 (17.2) | 22 (7.1) | 155 (50.3) | 23 (7.5) |
| I feel overwhelmed by the responsibility of being a parent | 75 (24.4) | 80 (25.9) | 11 (3.6) | 134 (43.5) | 8 (2.6) |
| Having child(ren) has meant having too few choices and too little control over my life | 62 (20.1) | 90 (29.2) | 13 (4.2) | 136 (44.2) | 7 (2.3) |
| I am satisfied as a parent* | 49 (15.9) | 28 (9.1) | 158 (51.3) | 62 (20.1) | 11 (3.6) |
| I find my child(ren) Enjoyable* | 8 (2.6) | 34 (11) | 11 (3.6) | 181 (58.8) | 74 (24) |

**\*-** Items reversed in scoring.

Specifically, mothers experienced an average increase of 2.51 points in parental stress scores compared to fathers ($\beta = 2.51$, 95% CI = 0.16, 4.87). For each additional year in parental age, the stress score increased by 0.18 points ($\beta = 0.18$, 95% CI = 0.01, 0.36). Similarly, for each additional year in the child's age, the parental stress score rose by 0.81 points ($\beta = 0.81$, 95% CI = 0.42, 1.19). Parents with a previous child diagnosed with NTD reported stress levels that were 7.88 points higher ($\beta = 7.88$, 95% CI = 3.22, 12.53) compared to those without such a history. Furthermore, parents of children with a ventriculoperitoneal shunt experienced stress levels 4.66 points higher than those whose children did not have this intervention ($\beta = 4.66$, 95% CI = 2.31, 7.02) **"Table 4"**.

## Discussion

Because birth defects are a major cause of morbidity and mortality in children under five, knowing the birth prevalence of neural tube defects and the factors contributing to parental stress is important because it helps in creating targeted support and resources to better meet the mental health needs of parents. This study found that the birth prevalence of neural tube defects is 52 per 10,000 deliveries. Important factors linked to parental stress included being a mother, older parental age, the age of the child, a previous history of having children with NTD, and having a ventriculoperitoneal shunt in the child.

Our study reveals a birth prevalence of neural tube defects that exceeds the findings of a systematic review and meta-analysis in Africa, which reported 21.4 cases per 10,000 births [1], as well as the World Health Organization's estimate of 22 per 10,000 births in Ethiopia [19]. However, it is lower than a systematic review conducted in Ethiopia, which reported a pooled prevalence of 71.48 cases per 10,000 births [20]. These discrepancies may arise from various factors, including racial, geographical, nutritional, socioeconomic, and biological differences, as well as the inherent differences in study settings. Referral hospitals typically report higher rates than population-based studies, as many NTD cases are

**Table 4. Simple and multiple linear regression results of stress among parents with neural tube defect children who have a follow-up in DCSH and DBCSH, Ethiopia 2024 (n = 308).**

| Covariates | Simple linear regression | | Multiple linear regression | |
|---|---|---|---|---|
| | β(95%CI) | P-value | β(95%CI) | P-value |
| Parental role *(Father)* **Mother** | 3.39 (0.74, 6.04) | 0.012 | **2.52 (0.16, 4.87)** | **0.037** |
| Age of the parents (in years) | 0.32 (0.13, 0.51) | 0.001 | **0.18 (0.01, 0.36)** | **0.044** |
| Average income (ETB) | 0.002 (−0.00, 0.004) | 0.109 | −0.00 (−0.00, 0.001) | 0.652 |
| Residency *(Urban)* **Rural** | −2.19 (−4.76, 0.37) | 0.094 | −.78 (−3.24, 1.65) | 0.522 |
| Educational status of the mother *(Diploma and above)* **No formal education** | −6.16 (−9.34, −2.97) | 0.000 | −2.8 (−6.87, 1.27) | 0.176 |
| **Primary education** | −1.89 (−4.61, 0.83) | 0.136 | 1.46 (−1.95, 4.89) | 0. 398 |
| **High school education** | −5.07 (−8.01, −2.12) | 0.001 | −1.85 (−4.97, 1.27) | 0.245 |
| Educational status of the father *(Diploma and above)* **No formal education** | −2.73 (−6.29, 0.83) | 0.132 | −0.72 (−4.64, 3.21) | 0.732 |
| **Primary education** | −4.83 (−7.52, −2.14) | 0.000 | −2.89 (−6.01, 0.24) | 0.071 |
| **High school education** | −3.93 (−6.69, −1.17) | 0.005 | −2.15 (−5.06, 0.76) | 0.147 |
| Occupational status of the respondent *(Private employee)* **Governmental employee** | 2.71 (−0.15, 5.56) | 0.063 | −1.57 (−4.45, 1.33) | 0.288 |
| **Housewife** | −0.25 (−2.84, 2.35) | 0.852 | −0.76 (−3.21, 1.65) | 0.530 |
| **Other** | −3.39 (−10.3, 3.56) | 0.338 | −3.16 (−9.51, 3.15) | 0.325 |
| Previous NTD history *(NO)* **Yes** | 9.7 (4.44, 14.97) | 0.000 | **7.82 (3.11, 12.52)** | **0.001** |
| Age of child (in Years) | 1.00 (0.58, 1.42) | 0.000 | **0.81 (0.42, 1.19)** | **0.000** |
| Sex of the child *(male)* **Female** | −0.42 (−2.61, 1.76) | 0.930 | −0.19 (−2.18, 1.8) | 0.85 |
| Types of neural tube defect *(Encephalocele)* **Meningocele** | −0.52 (−6.14, 5.09) | 0.855 | −3.34 (−9.01, 2.31) | 0.245 |
| **Myelomeningocele** | 6.73 (1.74, 11.73) | 0.008 | 2.87 (−1.98, 7.74) | 0.246 |
| Able to ambulate *(Yes)* **No** | 4.29 (1.98, 6.61) | 0.000 | 0.93 (−1.61, 3.46) | 0.47 |
| Urinary incontinence *(No)* **Yes** | 3.58 (1.39, 5.76) | 0.001 | 1.60 (−1.56, 4.78) | 0.319 |
| Bowel dysfunction *(No)* **Yes** | 2.68 (0.46, 4.89) | 0.018 | 1.13 (−2.18, 4.45) | 0.502 |
| Ventricle-peritoneal Shunt *(No)* **Yes** | 7.02 (4.52, 9.52) | 0.000 | **4.66 (2.31, 7.02)** | **0.000** |
| Clean intermittent catheter *(No)* **Yes** | 5.98 (2.69, 9.28) | 0.000 | 1.64 (−1.71, 5.00) | 0.336 |

directed to these facilities for diagnosis and termination. Nonetheless, our findings align with previous reports from Ethiopia [1,3].

The findings of this study highlight that being a mother is closely linked to higher stress levels. This aligns with previous research [14,21], which indicates that the pressures of parenting arise from caregiving responsibilities, societal expectations, and emotional investment in children's well-being [14]. Mothers often serve as the primary caregivers, facing more illness-related situations and demands than other family members [22].

Additionally, older parents also tend to experience slightly increased stress when caring for a child with a neural tube defect. This finding was supported by a study conducted by Macias et al. [23], showing that older mothers report higher stress levels related to concerns for their child and medicolegal issues [23]. Furthermore, as maternal age increases, concerns about fertility and potential pregnancy complications also heighten [24].

Similarly, parental stress levels tend to increase as a child grows older. This is largely due to parents becoming more aware of their children's limitations and challenges. Such awareness can lead to fears about long-term care and concerns regarding the child's future quality of life [25]. As children with disabilities grow, the demands on their parents often increase due to new challenges in care and development. Unfortunately, support systems tend to shift away from families as children age [23]. This transition in support may overwhelm the parents and intensify stress and anxiety, especially for those with limited resources [26].

Parents who have previously experienced an NTD report significantly higher stress levels than those who have not. Previous studies indicate that parents of infants with NTDs face emotional and financial burdens, leading to stress about potential complications [14]. The fear of repeating past experiences can create additional stress, as parents worry about their child's health outcomes and the possibility of severe disabilities [25,27]. Discrimination and stigma surrounding disabilities can further exacerbate this stress [28].

The placement of a Ventriculoperitoneal shunt is another factor contributing to increased parental stress. The complexities of the procedure, potential shunt-related complications, and their implications for the child's health are significant concerns for caregivers [29]. Additionally, many parents lack adequate knowledge about the type of shunt, its settings, and signs of malfunction, which can heighten anxiety [30]. Complications such as obstruction and infection further add to parental anxiety, particularly when caregivers are uninformed about potential risks [31]. Lastly, economic, psychological, and social factors play crucial roles in shaping the challenges parents face [32]. Access to resources, community support, and mental health services can significantly impact how parents manage stress.

Finally, the following limitations are acknowledged in this study. Firstly, being institutionally based and focused on comprehensive specialized hospitals, which primarily serve as referral centers, may not accurately reflect the true population-level birth prevalence of NTDs, as it excludes home births and deliveries in lower-level health facilities, thereby potentially overestimating prevalence due to referral bias. Secondly, the cross-sectional design limits our ability to establish causal relationships or capture changes in parental stress over time. Thirdly, parental stress was assessed using self-reported data collected at a single point during follow-up, which may be subject to recall bias or influenced by transient emotional states, daily stressors, or the immediate context of the clinic visit. Importantly, the absence of a comparison group (e.g., parents of typically developing children) means our findings describe within-group associations rather than absolute differences in stress levels. Future longitudinal or mixed-methods studies could address these gaps. Finally, while the quantitative approach enabled the identification of key predictors of stress, it could not fully capture the depth of parents' lived experiences, emotional journeys, or sociocultural coping mechanisms. Future research would benefit from a mixed-methods design, combining validated stress scales with in-depth qualitative interviews or focus group discussions.

## Conclusion and recommendation

This study reveals that the birth prevalence of neural tube defects was 52 cases per 10,000 deliveries, which was higher than the WHO's estimate for Ethiopia and underscored a significant public health concern. Several factors contributing to increased parental stress were identified, including being a mother, older parental age, the child's age, a previous history of NTDs in siblings, and the presence of a ventriculoperitoneal shunt. To address these challenges, healthcare providers implement targeted counseling and support programs for mothers and older parents, enhance prenatal education focused on NTD prevention, and establish support networks for families with a history of NTDs. Additionally, providing resources and information about managing stress related to the presence of medical devices, such as ventriculoperitoneal shunts, can further help mitigate the psychological impact on affected families.

## Supporting information

**S1 File. Birth prevalence and parental stress ass NTD Data.**
(DTA)

## Acknowledgments

We would like to thank the staff members of the Maternal and Child Health case team at DCSH and DBCSH, as well as all the data collectors, for their invaluable contributions to this study. Lastly, we are deeply appreciative of the parents and their newborns who willingly participated in our research.

## Author contributions

**Conceptualization:** Hailemariam Gezie, Fekadeselassie Belege Getaneh.

**Data curation:** Hailemariam Gezie, Muluken Amare Wudu, Fekadeselassie Belege Getaneh.

**Funding acquisition:** Hailemariam Gezie, Habtam Gelaye.

**Investigation:** Endalk Birrie Wondifraw, Habtam Gelaye, Fekadeselassie Belege Getaneh.

**Methodology:** Hailemariam Gezie, Habtam Gelaye, Fekadeselassie Belege Getaneh.

**Project administration:** Muluken Amare Wudu, Fekadeselassie Belege Getaneh.

**Resources:** Hailemariam Gezie.

**Supervision:** Endalk Birrie Wondifraw, Habtam Gelaye, Fekadeselassie Belege Getaneh.

**Validation:** Fekadeselassie Belege Getaneh.

**Visualization:** Fekadeselassie Belege Getaneh.

**Writing – original draft:** Muluken Amare Wudu, Fekadeselassie Belege Getaneh.

**Writing – review & editing:** Hailemariam Gezie, Endalk Birrie Wondifraw, Habtam Gelaye.

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
