## [Decision Letter · Decision Letter 0]

21 Sep 2025

Dear Dr. Getaneh,

Thank you for submitting your manuscript to PLOS ONE. After careful consideration, we feel that it has merit but does not fully meet PLOS ONE’s publication criteria as it currently stands. Therefore, we invite you to submit a revised version of the manuscript that addresses the points raised during the review process.

Please note that we have only been able to secure a single reviewer to assess your manuscript. We are issuing a decision on your manuscript at this point to prevent further delays in the evaluation of your manuscript. Please be aware that the editor who handles your revised manuscript might find it necessary to invite additional reviewers to assess this work once the revised manuscript is submitted. However, we will aim to proceed on the basis of this single review if possible. 

We look forward to receiving your revised manuscript.

Kind regards,

Jianhong Zhou

Staff Editor

PLOS ONE

Journal Requirements:

Reviewers' comments:

Reviewer's Responses to Questions

**Comments to the Author**

1. Is the manuscript technically sound, and do the data support the conclusions?

Reviewer #1: Partly

2. Has the statistical analysis been performed appropriately and rigorously?

Reviewer #1: No

3. Have the authors made all data underlying the findings in their manuscript fully available?

Reviewer #1: Yes

4. Is the manuscript presented in an intelligible fashion and written in standard English?

Reviewer #1: Yes

Reviewer #1: Congenital anomalies are understudied conditions especially in LMICs, so while your study is really commendable in terms of what you set out to do, the methodology lacks the required rigor.

1. The aim of the study was partly to provide a birth prevalence of NTD in the Amhara regional state, hence some background information of the catchment area of these hospitals is necessary to put the prevalence in context. For example, who delivers at this hospital, what proportion of the total deliveries of the region happen here? Are these mostly complicated cases who are referred?

2. Further information on who diagnosed the NTD at birth? Was it by midwives / nurses / pediatricians other trained professionals?

3. The authors mention both live and stillbirths were included. Providing separate prevalence rates by types of NTDs (separate for live and stillbirths) would give a clearer picture as anencephaly is a fatal condition while spina bifida may not be.

4. The other part the authors have aimed for is the psychological impact on parents. A qualitative study design would be most appropriate here. That would have really brought out the impact that such children have on their parents and families.

5. The measured stress levels are among those parents who are coming for follow-up. As there is no comparison group, just noting the stress levels simply gives a snapshot but a lot of nuances are lost.

6. Caring for children with such conditions is known to put a lot of strain on the parents and care-givers, but in this context there is no comparison to know whether stress is most at the time of diagnosis, does it reduce with time and support that they seem to be receiving since they are coming for the follow-up visits or does having to come for the follow-up actually put more strain on them.

7. Was sex of the affected child not used as a covariate in the analysis?

8. The distribution of NTDs varied between the hospitals, with 32 cases (0.55%) reported at DCSH and 19 cases (0.49%) at DBCSH – percentage-wise the numbers should be 63 & 37 % or kindly mention the word prevalence in the sentence.

**Do you want your identity to be public for this peer review?** For information about this choice, including consent withdrawal, please see our Privacy Policy

Reviewer #1: No

---

## [Author Response · Author response to Decision Letter 1]

17 Oct 2025

Point-by-Point response

We would like to express our deepest appreciation to the editors for their diligent management of the peer review process. Their careful oversight and guidance have been instrumental in ensuring the integrity and quality of our work. We would also like to extend our heartfelt gratitude to the esteemed reviewers who invested their valuable time and expertise in reviewing our paper. We sincerely appreciate the reviewer’s thoughtful and constructive feedback, which has helped us strengthen the clarity and contextual relevance of our study. Below are our point-by-point responses to the comments:

Response to Reviewer #1

Congenital anomalies are understudied conditions especially in LMICs, so while your study is really commendable in terms of what you set out to do, the methodology lacks the required rigor.

1. The aim of the study was partly to provide a birth prevalence of NTD in the Amhara regional state, hence some background information of the catchment area of these hospitals is necessary to put the prevalence in context. For example, who delivers at this hospital, and what proportion of the total deliveries of the region happen here? Are these mostly complicated cases that are referred?

Response:

We acknowledge this important point. Dessie Comprehensive Specialized Hospital (DCSH) and Debre Berhan Comprehensive Specialized Hospital (DBCSH) serve as tertiary referral centers for the Eastern Amhara region and neighboring zones (including parts of Oromia and Afar). While precise regional delivery statistics are not routinely published, these hospitals handle a substantial proportion of high-risk and complicated deliveries in their catchment areas due to their specialized maternal and neonatal services. As noted in the limitations section, our institution-based design may overrepresent referred or complicated cases, which could influence the observed prevalence. We have now added a brief clarification in the Methods (Study Area) section better to contextualize the hospitals’ roles and potential referral bias.

2. Further information on who diagnosed the NTD at birth? Was it by midwives/nurses/pediatricians, or other trained professionals?

Response:

Diagnoses of neural tube defects (NTDs) at birth were primarily made by attending obstetricians and residents during physical examination of the newborn, supported, when necessary, by neonatal imaging (e.g., cranial or spinal ultrasound). In cases of stillbirth or obvious external anomalies (e.g., myelomeningocele, encephalocele), diagnoses were confirmed by senior midwives in consultation with physicians. We have now clarified this in the Methods section.

3. The authors mention both live and stillbirths were included. Providing separate prevalence rates by types of NTDs (separate for live and stillbirths) would give a clearer picture as anencephaly is a fatal condition while spina bifida may not be.

Response:

This is a valuable suggestion. In our dataset, of the 51 NTD cases identified, 34 were live births and 17 were stillbirths. Anencephaly and complex/multiple defects accounted for all stillbirths (17/17). We have now included this stratification in the Results section (under “Birth prevalence of neural tube defect”).

4. The other part that the authors have aimed for is the psychological impact on parents. A qualitative study design would be most appropriate here. That would have really brought out the impact that such children have on their parents and families.

Response:

We agree that qualitative methods could offer rich, in-depth insights into parental experiences. However, our objective was to quantify the magnitude of parental stress and identify statistically significant predictors using a validated scale (the 18-item Parental Stress Scale), which is well-established in similar settings. While a mixed-methods approach would be ideal, resource and time constraints limited us to a cross-sectional quantitative design. We now acknowledge this limitation more explicitly and suggest mixed-method studies in the Discussion.

5. The measured stress levels are among those of parents who are coming for follow-up. As there is no comparison group, just noting the stress levels simply gives a snapshot, but a lot of nuances are lost.

Response:

We fully agree. Our study was not designed to compare stress levels between parents of children with NTDs and those of typically developing children, nor to track changes over time. Instead, our focus was on identifying factors associated with higher stress within this vulnerable group to inform targeted interventions. We have clarified this scope in both the Introduction and Limitations sections and emphasized that findings should be interpreted as within-group associations rather than absolute comparisons.

6. Caring for children with such conditions is known to put a lot of strain on the parents and care-givers, but in this context there is no comparison to know whether stress is most at the time of diagnosis, does it reduce with time and support that they seem to be receiving since they are coming for the follow-up visits or does having to come for the follow-up actually put more strain on them.

Response:

This is an insightful observation. Our cross-sectional design captures stress at a single point during follow-up, not at diagnosis or longitudinally. However, our regression analysis shows that parental stress increases with the child’s age (β = 0.81, p < 0.001), suggesting that stress may accumulate over time rather than diminish, even among families engaged in care. We discussed the possible reason for this in the discussion section.

7. Was the sex of the affected child not used as a covariate in the analysis?

Response:

Yes, the child’s sex was initially considered as a covariate in the simple linear regression. However, it was not statistically significant (p > 0.25) and was therefore excluded from the final multiple regression model (only variables with p < 0.25 in bivariate analysis were retained for multivariable modeling).

We acknowledge that this analysis was inadvertently omitted from the original version of Table 4. In the revised manuscript, we have now added the simple linear regression result for the child’s sex to Table 4 for transparency and completeness.

8. The distribution of NTDs varied between the hospitals, with 32 cases (0.55%) reported at DCSH and 19 cases (0.49%) at DBCSH – percentage-wise the numbers should be 63 & 37 % or kindly mention the word prevalence in the sentence.

Response:

Thank you for this correction. The values 0.55% and 0.49% refer to the birth prevalence (i.e., proportion of total deliveries) at each hospital, not the percentage distribution of total NTD cases. To avoid confusion, we have revised the result section.

---

## [Decision Letter · Decision Letter 1]

14 Nov 2025

Birth prevalence and parental stress associated with neural tube defects in Amhara’s public comprehensive specialized hospitals, Ethiopia, 2024.

PONE-D-25-10943R1

Dear Dr. Getaneh,

We’re pleased to inform you that your manuscript has been judged scientifically suitable for publication and will be formally accepted for publication once it meets all outstanding technical requirements.

Kind regards,

James J Cray Jr., Ph.D.

Academic Editor

PLOS ONE

Additional Editor Comments (optional):

Reviewers' comments:

Reviewer's Responses to Questions

**Comments to the Author**

Reviewer #1: All comments have been addressed

2. Is the manuscript technically sound, and do the data support the conclusions?

Reviewer #1: Yes

3. Has the statistical analysis been performed appropriately and rigorously?

Reviewer #1: Yes

4. Have the authors made all data underlying the findings in their manuscript fully available?

Reviewer #1: Yes

5. Is the manuscript presented in an intelligible fashion and written in standard English?

Reviewer #1: Yes

Reviewer #1: All comments have been addressed satisfactorily and the required modifications have been made to the manuscript. I have no further comments.

**Do you want your identity to be public for this peer review?** For information about this choice, including consent withdrawal, please see our Privacy Policy

Reviewer #1: No

---

## [Editor Report · Acceptance letter]

PONE-D-25-10943R1

PLOS ONE

Dear Dr. Getaneh,

I'm pleased to inform you that your manuscript has been deemed suitable for publication in PLOS ONE. Congratulations! Your manuscript is now being handed over to our production team.

Kind regards,

on behalf of

Dr. James J Cray Jr.

Academic Editor

PLOS ONE